# Does a specialized assessment improve vocational outcomes for people on sick leave with a suspected common mental disorder? Results from the Mental Health Assessment Study (MeHAS)

**Andreas Hoff** [1] *, **Anders Bo Bojesen** [1], **Jonas Fisker** [2], **Rie Mandrup Poulsen** [3], **Carsten Hjorthøj** [1,4], **Merete Nordentoft** [1], **Lene Falgaard Eplov** [1]

**1** CORE (Copenhagen Research Centre for Mental Health), Hellerup, Denmark, **2** Hejmdal Private Hospital, Copenhagen, Denmark, **3** The National Board of Social Services, Odense, Denmark, **4** Department of Public Health, Section of Epidemiology, University of Copenhagen, Copenhagen, Denmark

* doktorhoff@gmail.com

**Data Availability Statement:** Data is not publicly available due to patient confidentiality and ethical

## Abstract

### Background

In Denmark, 50% of those on long-term sick leave are affected by common mental disorders (CMDs), and it has been argued that detection in primary care has been insufficient. The Mental Health Assessment Study (MeHAS) assesses if specialized mental health assessments can enhance return to work for this group. This study aimed to estimate the effect of a specialized mental health assessment for people on sick leave with a mental health disorder, on return to work and mental health care utilization.

### Methods and findings

In this experimental study, sickness absentees were referred from a sick leave benefit management agency. Before intervention allocation, they had already received a standard health assessment in general practice. The intervention group received an additional specialized mental health assessment, while the control group did not. We compared the groups on several vocational outcome measures, the primary being proportion in work after one year. Other outcomes were weeks in work, time to return to work (RTW) and different measures of service utilization. We included 717 in the intervention group and 756 in the control group. On the primary outcome, proportion in work, we observed no differences between the groups at 12 months (53.9% vs. 58.7% in the control group, p = 0.133). Moreover, after one year, the control group showed faster RTW at 12-month follow-up (HR 0.79, p<0.001) and 3.1 more weeks in work (p<0.001). In the intervention group, participants received more hospital-based outpatient mental healthcare.

restrictions imposed by the Capital Region Psychiatric Legal Research Team. Data access can be requested through contact to the Capital Region Psychiatric Legal Research Team via email at forskningsjura.rigshospitalet@regionh.dk.

**Funding:** Funding was received from the Danish Agency of Labour Market and Recruitment to conduct the study, and by contributions from the collaborating Danish municipalities who delivered the vocational rehabilitation interventions, no grant number was given by the funding party, but funding was given to AE and LE. None of these funding organizations had any influence on the interpretation of results, or the writing or reviewing of the manuscript; further, no funding organization saw the manuscript before submission. The collaborators contributed to developing the study protocol, which included population definition and determining outcomes.

**Competing interests:** The authors have declared that no competing interests exist.

## Conclusion

Providing a specialized mental health assessment was associated with fewer weeks in work and longer sick leave duration (secondary outcomes), but the proportion in work at 12-month follow-up (primary outcome) did not differ between the groups. The intervention was associated with a higher likelihood of receiving specialized mental healthcare services, perhaps because more needs were met. Given the substantial risk of selection bias, results should be treated with caution.

## Introduction

At any given time, as much as almost 20% of the adult population fulfil the criteria for a common mental disorder (CMD) such as depression, anxiety, distress, adjustment disorder or exhaustion disorder [1], and in Denmark 50% of those on long-term sick leave have a CMD [2]. Yet, many mental disorders go undetected and therefore untreated [3]. For example, one study showed that 21% of people with long-term sickness absence had an undetected mental health disorder [4]. In Denmark, treatment as usual for people with CMD during sick leave includes a health assessment in general practice, which is communicated to the vocational rehabilitation service provider. Improving this assessment has yielded mixed results, with studies showing positive vocational effects in some groups [5, 6] but no effect in others [5, 7]. One negative study did however find that improving the mental health assessment had positive health economic effects in the form of lower utilization of hospital-based mental health services [7], and another study found lower symptom levels in the group of people with sub-threshold mental health symptoms [5].

*The Mental Health Assessment Study* (MeHAS) is one of several studies in the IBBIS Project. As part of the project, we conducted two randomized controlled trials (RCT) that are reported elsewhere [8, 9]. The control groups received service as usual after an eligibility assessment at baseline that included the specialized IBBIS Mental Health Assessment (IBBIS-MHA). Since such an assessment may in itself impact vocational outcomes, as described above, the objective of the MeHAS study was to study the impact of the IBBIS-MHA. We hypothesized that the IBBIS-MHA intervention would improve vocational outcomes.

To sum up, the rationale for this study was two-fold: 1) to evaluate the external validity of our two RCTs by determining the degree to which the outcomes of the service as usual group were due to the mental health assessment per se; and 2) to contribute to the body of research on the impact of improved mental health assessments.

## Methods

The study was pre-registered on www.osf.io (doi: 10.17605/osf.io/7tcaf).

### Study design and participants

We compared two groups of long-term sickness absentees that were allocated to different interventions. Participants were recruited from April 2016 to April 2018. Eligibility was determined by staff in the Copenhagen jobcentre, which is the local public office managing payment of sick leave benefits during sickness absence. Inclusion criteria were: 1) being on sick leave from work for more than four weeks with a suspected or established mental health disorder; 2) receiving sickness benefit from Copenhagen Municipality; and 3) being at least 18 years of age.

Exclusion criteria were: 1) showing aggressiveness or other adverse behaviour and 2) living outside the Copenhagen Municipality.

## Allocation

Fig 1 displays the flow of study participants. In general, all incident long-term sickness absentees in Copenhagen Municipality are scheduled to meet their case manager in the Jobcentre; the Jobcentre in any Danish municipality is the institution responsible for providing sick-leave benefit for citizens on sick-leave (absentees). Every week during the study inclusion period, a list was prepared by a staff member in the municipality administration, including all the absentees deemed eligible for referral to this study, since the Jobcentre collects information about reason for sick-leave, age, first date and duration of sick-leave, which was information sufficient to establish study eligibility. Absentees were included in the list if a mental health disorder was suspected or established as a main cause of sick leave on the basis of a mental health assessment performed by the absentee's general practitioner or, in some cases, information provided by the absentee.

From said list, an administrative staff member selected a subset of individuals to be approached to be offered inclusion to the study's experimental intervention group. The size of this subset corresponded to the assessment capacity in the following week, since this number varied over time. Allocation to the experimental condition only occurred if the absentee provided written consent in accordance with the Helsinki Declaration. Each potential participant was informed that the intervention was an experimental intervention that differed from the standard assessment provided to everybody else. The remaining subset of absentees, who were not offered inclusion in the intervention group, received service as usual and were allocated to the control group. These participants did not receive any experimental treatment and could therefore be observed without consent, in accordance with Danish legislation. Staff members were supposed to know as little as possible about the absentees in order to perform the allocation randomly. After the intervention—the IBBIS Mental Health Assessment (IBBIS-MHA)—eligible participants were offered randomization to one of two RCTs. Randomized participants were excluded from this study, unless they were allocated to service as usual in the RCTs; see Fig 1. This exclusion ensured that only one condition—the IBBIS-MHA intervention—differed between the study groups.

## Interventions

Before allocation to either the control group or the intervention group, most eligible sickness absentees had been assessed by their general practitioner. In the intervention group, the IBBIS-MHA was added to the usual health assessment in general practice.

**Control group.**  The control group, receiving service as usual, received health care provided by their general practitioners and occasionally a psychiatrist or other clinicians to whom the general practitioner may have referred. Service as usual included vocational rehabilitation in the jobcentres. Vocational rehabilitation consisted of occasional work function assessments, various programs supporting graded RTW and courses aiming at symptom and work management.

**Intervention group.**  In the intervention group, the IBBIS-MHA was performed by a psychiatrist or by a mental health professional (a psychologist, nurse, or psychiatric medical resident) supervised by a psychiatrist. Depending on their previous experience, assessors were trained and directly supervised until they were deemed capable of doing assessment interviews independently. The IBBIS-MHA consisted of a clinical interview focusing on current mental health and a systematic evaluation of mental health issues. The clinical interview started with

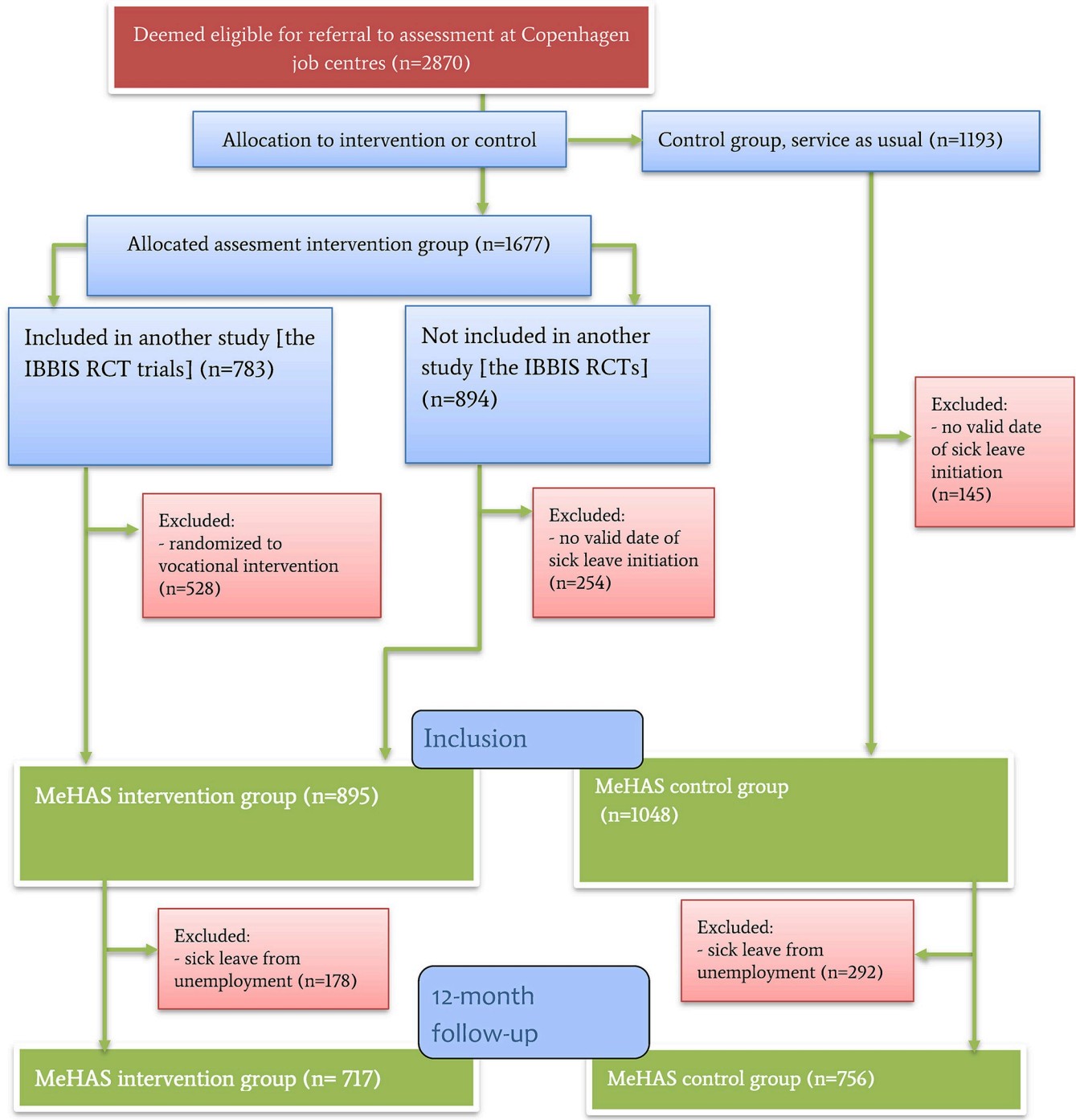

**Fig 1. Participant flow-chart; MeHAS: Mental Health Assessment Study; RCT: Randomized, controlled trial.**

an only slightly structured clinical interview where the participants was asked about main health issues, what they experinced as main cause(s) of sick leave. That was follow by 1) the semi-structured *MINI International Neuropsychiatric Interview* [10], to ensure a systematic aproach to assessment of main mental health symptom domains; furthermore, to screen for personality disorders it was followed by the clinician-rated 2) *Standardized Assessment of*

*Personality–Abbreviated Scale* (SAPAS) [11] and 3) *Attention deficit hyperactivity disorder symptom checklist for adults, Adult Self-Report Scale* (ASRS) [12]. The *Mini-Mental State Examination* (MMSE) was used if dementia was clinically suspected by the assessor [13]. Before the interview, participants answered the *Four Dimensional Symptom Questionnaire (4DSQ)* that measures levels of depression, anxiety, distress and somatization [14], and the assersors had access to the results in their clinical assessment. E.g., this questionnaire includes threshold values to guide the extent of symptoms of depression, anxiety, stress, and somatization, in order to establish what the main diagnosis should be. Assessment reports were sent to the benefit case managers, who used this information when handling each specific case. The plan was that participants allocated to the intervention group would receive the same vocational rehabilitation as the control group.

## Outcomes

**Primary and secondary outcomes.** The primary outcome at 12-month follow-up was proportion of study participants in full-time stable RTW, where such stability is defined as working for at least four consecutive weeks without receiving sick leave benefit. Secondary outcomes were 1) proportion of study participants in stable RTW, part-time or full-time, at 12-month follow-up; 2) total weeks in work (full-time) from baseline to 12-month follow-up; 3) total weeks in work (full-time or part-time) from baseline to 12-month follow-up; 4) time to either part-time or full-time stable RTW measured at 12-month follow-up; and 5) time to full-time stable RTW (defined as above) measured at 12-month follow-up.

**Explorative outcomes.** In each intervention group, we observed which vocational rehabilitation interventions were provided in the social sector, primary health care sector and hospital-based care (secondary) sector. Regarding social sector vocational rehabilitation, we described 1) the number of employment consultant meetings; 2) the number of employment consultant virtual contacts (mail, phone, etc.); and 3) the quantity of vocational rehabilitation services. Furthermore, we described the usage of general practitioners, subsidized psychologists, subsidized psychiatrists, and hospital-based psychiatric services.

**Ethics statement.** The IBBIS Trials was evaluated and approved by the Regional Ethics Committees of the Capital Region (# H-16015724) and the Danish Data Protection Agency (#RHP-2016-006). It was conducted in accordance with Danish and European regulations. An IBBIS team member informed every participant in the intervention group about the objective of the study and the implications of participation, and all participants gave oral and written consent before enrolment. Participants in the control group were not asked for consent as data on these were acquired through Danish Registers. According to Danish law no further approvals are required when conducting register-based studies. All data processing was conducted on Statistics Denmark's servers in accordance with EU and Danish legislation.

## Analyses

**Baseline.** Baseline was defined as the third week of sick leave, and participants had their first jobcentre meeting about benefits in week four at the earliest. Baseline variables were sex, age, education, civic status, employment, sick leave benefit period before baseline and general benefit usage history in the two years preceding baseline since these variables are known predictors of vocational outcomes [15]. For the intervention group, we described self-reported symptoms of depression, anxiety and distress, self-efficacy, life quality and a range of other measures. Furthermore, we included diagnoses obtained from the general practitioners and the IBBIS-MHA as well as scores on the scales *Standardized Assessment of Personality–Abbreviated Scale (SAPAS)* and *Attention deficit hyperactivity disorder symptom checklist for adults*

*(ASRS)*. The baseline characteristics of the treatment and control groups were described using two-way tables with counts and percentages for categorical variables and mean standard deviations (SD) and medians for numeric variables. Categorical variables were tested for differences using chi-square and numerical variables using the Wilcoxon rank-sum. Diagnoses based on the assessment were described per diagnostic ICD10 categories, and Sankey charts were used to visualize the relationship between diagnoses given in general practice and during the intervention assessment.

**Outcomes.** The primary outcome—proportion of participants in full-time stable employment at 12 months after baseline—was tested using logistic regression, as was the equivalent secondary outcome of proportion in part-time stable employment.

*Secondary outcomes.* Weeks in work (full-time and part-time) was analysed using linear regression with robust standard errors. Secondary time-to-event outcomes were tested using Cox regression. Supplementary rank-based tests of the time-to-event variables for all relevant participants were also conducted. Cox regression coefficients were reported as hazard ratio estimates, and logistic regression coefficients were also exponentiated to odds ratios. Linear regression estimates were differences in means and rate ratios. All regression estimates were adjusted for imbalanced variables at baseline (those with p-values below 0.05).

*Exploratory outcomes.* The groups' use of services was analysed using register data, observing the usage of general practice consultations, hospital-based mental health services, vocational rehabilitation, and employment consultations.

After allocation, some participants in the intervention group could be included in the IBBIS RCTs mentioned in the introduction above. These participants were randomized in a 1:1:1 ratio to one of three arms, and only the one-third allocated to the control groups was not excluded from this study. Participants randomized to one of the two experimental interventions in the RCTs were excluded from MeHAS, since these two groups received additional interventions that aimed at improving vocational outcomes. Since two-thirds of participants in this group were excluded, MeHAS intervention group participants who were randomized and included in the RCTs' SAU group were weighted with 1/3. All other participants were analysed with weight 1.

**Protocol deviations.** Post-hoc, we discovered a skewed distribution of vocational status before sick leave. As the imbalance was only found in people on sick leave from unemployment, we decided to exclude this group. We deemed balance in this variable to be crucial, as the IBBIS RCTs showed that employment status at baseline is a major explanatory variable in vocational outcomes [8, 9]. All deviations from the pre-registered protocol, including the important deviation described here, are exhaustively reported in S1 File. Post-hoc, we decided to conduct two sensitivity analyses. These are also described in S1 File.

## Results

### Baseline

We initially allocated 1677 participants to the intervention group and 1193 to the control group. In the intervention group, 783 were randomized to the IBBIS RCTs, and the two-thirds (n = 528) allocated to the two intervention groups in these RCTs were subsequently excluded from the MeHAS study. After allocation, we discovered a skewed distribution of vocational status before sick leave and therefore decided to exclude persons on sick leave from unemployment, since vocational status at initiation of sick leave influences vocational outcomes [8, 9]. This left 717 in the intervention group and 756 in the control group. The flow of participants is displayed in Fig 1, and the baseline description of the included population is seen in Table 1.

**Table 1. Baseline distribution of the analyzed groups (after exclusion of persons on sick leave from unemployment); BDI: Bech Depression Inventory; BAI: Bech Anxiety Inventory; PSS: Perceived Stress Scale; WSAS: Work and Social Adjustment Scale; 4DSQ: Four Dimensional Questionnaire; KES: Karolinska exhaustion disorder scale; IPQ: Illness Perception Questionnaire; EQ5DL: Health related quality of life; QoLs: Quality of Life Scale; RTW-SE: Return to work-self efficacy; SPS: Stepford Presenteeism scale; GSE: Generalized Self-Efficacy Scale; SAPAS: Assessment of Personality – Abbreviated Scale; ASRS: Attention deficit hyperactivity disorder symptom checklist for adults.**

| Variable | | MeHAS intervention | Control group | p |
|---|---|---|---|---|
| | n (%) | 717 (100) | 756 (100) | |
| | n (%, weighted) | 346 (100) | 756 (100) | |
| Sex | Female | 255 (73.9) | 530 (70.1) | 0.227 |
| | Male | 90 (26.1) | 226 (29.9) | |
| Age | Mean years (SD) | 40.25 (10.52) | 40.3 (10.38) | 0.868 |
| Education | 1. primary | 73 (21.1) | 170 (22.5) | 0.49 |
| | 2. secondary and vocational | 101 (29.2) | 203 (26.9) | |
| | 3. prof./academic | 68 (19.7) | 174 (23) | |
| | n/a | 104 (30.1) | 209 (27.6) | |
| Civic status | 1. Not married/cohabitating | 188 (54.3) | 368 (48.7) | 0.09 |
| | 2. Married/cohabitating | 158 (45.7) | 388 (51.3) | |
| Weeks with sickness benefit before baseline | | 7.78 (10.98) | 6.72 (8.82) | 0.717 |
| Weeks with any benefit and no salary before baseline | | 9.28 (18.69) | 7.5 (16.34) | 0.08 |
| Employment before sick leave, n (%) | | 346 (100) | 756 (100) | |
| BAI (anxiety) | | 19 (9.83) | n/a | |
| BDI (depression) | | 24.25 (10.43) | n/a | |
| WSAS (functioning) | | 23.21 (8.42) | n/a | |
| PSS (distress) | | 24.37 (6.1) | n/a | |
| IPQ (self-efficacy) | | 15.05 (3.62) | n/a | |
| QoLs (life quality) | | 62.08 (12.91) | n/a | |
| KES (exhaustion) | | 82.82 (15.06) | n/a | |
| GSS (self-efficacy) | | 23.6 (6.61) | n/a | |
| 4DSQ (somatization) | | 12.73 (6.98) | n/a | |
| 4DSQ (subscale: distress) | | 19.36 (7.62) | n/a | |
| 4DSQ (subscale: anxiety) | | 6.57 (5.8) | n/a | |
| 4DSQ (subscale: depression) | | 3.24 (3.32) | n/a | |
| RTW-SE (RTW self-efficacy) | | 13.85 (7.76) | n/a | |
| EQ5 (life quality) | | 0.69 (0.15) | n/a | |
| ASRS (ADHD symptoms) | | 7.62 (4.19) | n/a | |
| SAPAS (personality disorder screening items positive) | | 2.74 (1.23) | n/a | |

In the analysed population, the mean age was 40.25 (SD 10.52) and 40.3 years (SD 10.38) respectively, and most participants were female. Equally many were married or cohabiting, and all educational levels were roughly equally represented. During the two years preceding baseline, the number of weeks with any social benefit and no salary was 9.28 (SD 18.69) in the intervention group and 7.5 (SD 16.34) in the control group, including the three weeks in the index period before inclusion (p = 0.08 for difference). The intervention group showed anxiety levels on BAI 19 (SD 9.83) in the middle of the moderate interval (16 to 25) and baseline depression symptom levels on BDI of 23.91 (10.3), indicating moderate to severe depression (scores from 19 to 29). Baseline values of included participants before exclusion of unemployed participants are found in S2 File.

Data on the distribution of diagnosis at referral vs. after the assessment are found in S2 File.

**Table 2. Vocational outcomes at 12-month follow-up.** SD: Standard deviation; CI: 95% confidence interval; OR: Odds ratio; HR: Hazard ratio; RR: Rate ratio.

| | Group values | | Group comparisons | |
|---|---|---|---|---|
| | **Control group** | **Intervention group** | **Estimate** | **p-val.** |
| n (%) | 756 (100) | 717 (100) | | |
| Full-time work status at 12 months n (%) | 444 (58.7) | 186 (53.9) | 0.82 [OR]<br>(CI: 0.64 to 1.06) | 0.133 |
| Part- or full-time work status, 12 months, n (%) | 458 (60.6) | 194 (56.1) | 0.83 [OR]<br>(CI: 0.64 to 1.07) | 0.154 |
| Weeks in full time work during 12 months (SD) [median] | 21.3 (16.5) [23] | 18.1 (14.8) [19] | -3.12 [$\delta_{weeks}$]<br>0.85 [RR]<br>(CI: 0.78 to 0.93) | <0.001 |
| Weeks in part- or full-time work during 12 months (SD) [median] | 28.0 (20.1) [33.5] | 23.0 (18.6) [24] | -5.02 [$_{weeks}$]<br>0.82 [RR]<br>(CI: 0.76 to 0.90) | <0.001 |
| Sick leave duration before full-time work, days (SD) [median] | 170.8 (118.9) [140] | 181.8 (119.6) [175] | 0.88 [HR]<br>(CI: 0.78 to 1.00) | 0.053 |
| Sick leave duration before part- or full-time work, days (SD) [median] | 120.8 (127.9) [63] | 148.3 (132.5) [98] | 0.79 [HR]<br>(CI: 0.70 to 0.89) | <0.001 |

## Primary and secondary outcomes

On the primary outcome—proportion in full-time stable RTW at 12-month follow-up—we saw no difference between the groups. Secondary outcomes: A similar result was seen on the secondary outcome, proportion in part- or full-time stable RTW at 12-month follow-up The intervention group had fewer weeks in work (both full-time and part-time work) between baseline and 12-month follow-up (full-time: -3.1 weeks, 95%CI: -4.8 to -1.4, p<0.001; part-time: -5 weeks, 95%CI: -7.1 to -2.8, p<0.001). In terms of sick leave duration, we saw no difference using the full-time definition (140 vs 175 median days: HR 0.88, 95%CI: 0.76 to 1.00, p = 0.053) but using the part- or full-time definition, the control group showed faster RTW (63 vs 98 median days: HR 0.79, 95%CI: 0.70 to 0.89, p<0.001). All vocational outcomes are displayed in Table 2, and graphs A through D in Fig 2 display Kaplan-Meier curves of the full-time (A) vs. at least part-time (B) criterion and proportion of unemployed after baseline, using the full-time (C) vs. the at least part-time (D) criterion.

## Exploratory outcomes

The intervention group received more services on a range of measures, including a higher number of sessions at their general practitioners and more meetings with their employment consultants (physical as well as virtual meetings). The average level of vocational rehabilitation services was higher (both in terms of course duration and total number of hours); see Table 3.

Post-hoc analyses showed no substantial deviations from main analyses, and results are found in S2 File.

## Discussion

### Main findings

This study aimed to improve the RTW process by introducing an additional specialized mental health assessment for people on sick leave due to a mental health condition. This group was compared to a control group of similar participants who did not receive the intervention. We tested for statistical differences among the groups, and to our surprise found that allocation to the intervention was associated with longer duration of sick leave before part-time RTW at

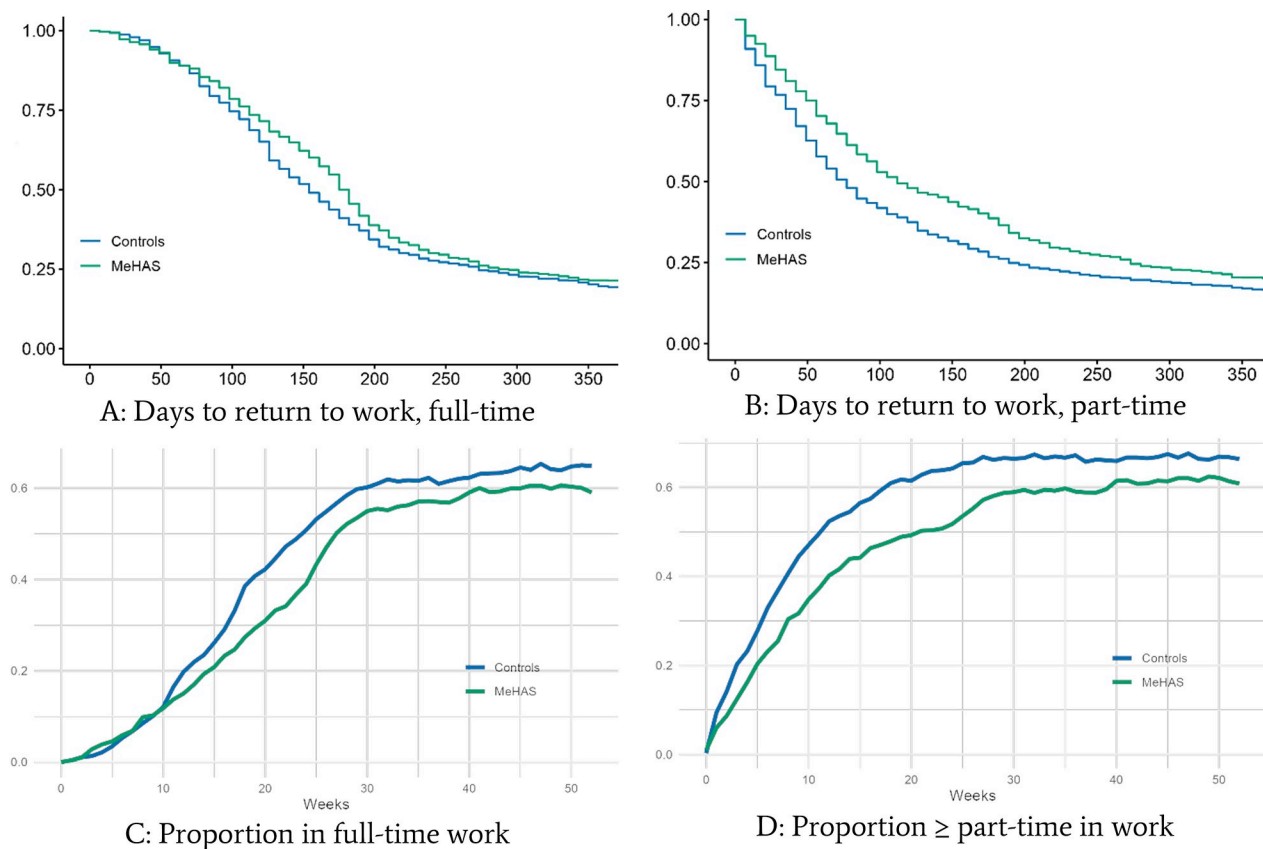

**Fig 2. Curves of vocational outcomes.** A: Kaplan-Meier curve displaying Time to return-to-work, full-time; B: Kaplan-Meier curve displaying time to return to work, part- or full-time; C: Proportion not in full-time work; D: Proportion not in at least part-time work.

12-month follow-up. It was also surprising that the intervention group had fewer weeks in work at 12-month follow-up, no matter whether RTW was defined as full-time or at least part-time (the difference being three and five weeks less, respectively). In addition, we found that the specialized assessment discovered previously undetected depression and anxiety. Further-more, the intervention led to higher utilization of services, meaning more meetings with the employment consultant as well as the general practitioner and longer duration and quantity of vocational rehabilitations courses.

## Interpretation of findings

During the one-year follow-up period, the intervention group generally worked less but received more services. When RTW was defined as either part- or full-time, the differences were greater than when using only the full-time definition. Hence, the control group displayed more graded RTW.

One possible explanation could be that the intervention delayed vocational rehabilitation service delivery, which may again have led to delayed RTW in the intervention group. In the control group, participants did not receive further assessment at their first meeting in the job-centre, and hence their vocational rehabilitation could start at the time of allocation. In the intervention group, when participants were allocated, no vocational rehabilitation was sched-uled before the results of the intervention assessment were available to the sick leave benefit manager. From the implementation studies of the IBBIS RCTs, we know that this process

**Table 3. Interventions delivered from Baseline to 12-month follow-up; GP: General Practitioner; VR: Vocational Rehabilitation; EC: Employment Consultant; SD: Standard Deviation; ER: Emergency Room; n/a: not available.**

| Service type | Specific intervention measure | | MeHAS-intervention | Control group | p-value |
|---|---|---|---|---|---|
| *Register data: Mental health care* | Sessions, GP | Mean (SD); Median | 8.42 (5.6); 7 | 7.65 (5.3); 6 | 0.023 |
| | Sessions, psychologist | Mean (SD); Median | 0.81 (2.5); 0 | 0.61 (2.2); 0 | 0.153 |
| | Sessions, psychiatrist | Mean (SD); Median | 0.40 (1.7); 0 | 0.36 (1.5); 0 | 0.569 |
| | ≥ 1 out-patient psych. contact | n (proportion, [%]) | 62 (18.0) | 99 (13.1) | 0.039 |
| | ≥ 1 psych. admission | n (proportion, [%]) | 33 (9.5) | 85 (11.2) | 0.459 |
| | ≥ 1 psych. ER contact | n (proportion, [%]) | 0 (0.0) | 0 (0.0) | N/A |
| *Register data: Vocational rehabilitation* | EC meetings | Mean (SD); Median | 2.67 (1.9); 2 | 1.92 (1.9); 1 | 0.000 |
| | EC virtual contacts | Mean (SD); Median | 1.73 (1.7); 1 | 2.08 (1.7); 2 | 0.001 |
| | VR course | n (proportion [%]) | 87 (25.1) | 125 (16.5) | 0.001 |
| | VR course, hours (cumulated) | Mean (SD); Median | 13.1 (33.5); 0 | 8.13 (24.0); 0 | 0.001 |
| | VR course duration, days (start-to-end) | Mean (SD); Median | 13.5 (29.2); 0 | 8.53 (23.9); 0 | 0.001 |

usually took approx. three weeks. Different hazards in the first three weeks could not explain the differences since the sensitivity analysis excluding this period did not substantially alter the results.

Another possible explanation could be that the assessment in the intervention group led to the discovery of previously undetected depression and anxiety that warranted treatment, and that the waiting time for treatment elongated sick leave.

A third explanation could be that the intervention led to more conservative assessment conclusions regarding prognosis. From studies of implementation of vocational goals in mental healthcare, we know that it is difficult to get health care staff to think in terms of vocational rehabilitation [16]. The fact that the assessments were performed by mental healthcare professionals employed and trained in hospital-based psychiatry could have affected prognosis. A process evaluation of the IBBIS RCTs has shown that mental healthcare staff in these studies sometimes advocated for a longer sick leave benefit period [17].

**Comparison with similar studies.** Another somewhat similar study showed no vocational impact of a specialized mental health assessment [5]: The improved assessment in that study was performed by a psychiatric specialist with many years of experience, compared to only the assessments from the general practitioners in the control group. In our study, the assessors came from diverse professions, and a large part of the assessments were performed by healthcare persons with varying diagnostic experience. This may have negatively impacted the quality of the assessment compared to the already mentioned study. Another study, of the effects of hastened assessment by a psychiatrist demonstrated no impact on sick leave benefits but lowered healthcare utilization costs [18]. That was observed when compared to a control group who did not receive the hastened specialized psychiatric assessment but only the usual assessment at general practitioners that was delivered in both groups. That goes against the tendency towards higher service utilization seen in our study. These studies are also Danish, and this fact somewhat reduces their applicability in an international context.

## Strengths and limitations

This study has several strengths. It was pre-registered, and all outcomes were exhaustively reported. Analysis was only commenced when all data had been collected; the pre-registered statistical analysis plan was followed, and all deviations were reported with explanations provided. We used quasi-randomization to form a control group, and we used register data to calculate outcomes. Hence, degree of missing data was very low.

The study is limited by the risk of selection bias. In the referring organization handling primarily unemployed sick leave benefit recipients, the intended random allocation procedure was not followed. Hence, we had to exclude these participants. In the remaining referral agencies, we also saw signs of selection bias, and we cannot assert that we fully managed to adjust for its effects. The study is also limited by the fact that referral to the study may have been biased, since the responsible staff tended to only refer those sickness absenteesif who were likely to be enrolled in an RCT, even though the manuals stipulated that all absentees should be allocated if any mental health condition was suspected. Furthermore, the sample size was a consequence of properties in other studies [8, 9] and did not fully support the testing of the hypotheses of this study.

## Implications for policy and further research

Our data does not support that our intervention should be routinely implemented, in order to enhance vocational outcomes during sick leave. Conversely, we saw indications that our intervention may have obstructed the return-to-work process, why further research should be conducted to clarify the effects of mental health assessment. In order to do so, it is important to deliver any such study intervention readily after study group allocation, in order to make sure that the effect of any difference between study groups cannot be merely attributed to the effects of intervention delay or different timing in intervention delivery between groups. Furthermore, such study should utilize a rigorous randomization procedure to avoid any selection bias. Preferably, the inclusion criteria for this group should should still be rather inclusive. as in this study, in order to avoid risk of low generalizability to real-world sickness absentees.

## Conclusion

We examined whether a specialized mental health assessment would impact vocational outcomes and found that this intervention was associated with lower work participation on several, but not all, outcome measures. Yet, the fact that the allocation was not entirely random may have introduced bias and substantial uncertainty about the validity of the results. Furthermore, since vocational outcomes are quite sensitive to contextual factors like sick leave benefit legislation, our findings might not be replicable in other national contexts. Nevertheless, interventions may unnecessarily delay recovery if they are commenced too late. Necessary assessments should therefore be provided as soon as possible, without any avoidable delay. However, more studies are needed.

## Supporting information

**S1 File. Protocol deviations.**
(DOCX)

**S2 File. Results.**
(DOCX)

## Acknowledgments

Recruitment and referral of participants was aided by the administrative staff in Copenhagen Municipality's vocational rehabilitation agencies. We are most thankful for the participants' willingness to be included in the study, and for the support of the collaborating organizations. All authors declare no conflicts of interest.

## Author Contributions

**Conceptualization:** Lene Falgaard Eplov.

**Data curation:** Andreas Hoff.

**Formal analysis:** Andreas Hoff, Anders Bo Bojesen.

**Funding acquisition:** Lene Falgaard Eplov.

**Investigation:** Andreas Hoff.

**Methodology:** Andreas Hoff, Anders Bo Bojesen.

**Project administration:** Andreas Hoff.

**Supervision:** Anders Bo Bojesen, Merete Nordentoft.

**Writing – original draft:** Andreas Hoff.

**Writing – review & editing:** Andreas Hoff, Anders Bo Bojesen, Jonas Fisker, Rie Mandrup Poulsen, Carsten Hjorthøj, Merete Nordentoft, Lene Falgaard Eplov.

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
