## [Decision Letter · Decision Letter 0]

29 Dec 2023

PMEN-D-23-00014

Does a specialized assessment improve vocational outcomes for people on sick leave with a suspected common mental disorder? Results from the Mental Health Assessment Study (MeHAS)

PLOS Mental Health

Dear Dr. Hoff,

Thank you for submitting your manuscript to PLOS Mental Health. After careful consideration, we feel that it has merit but does not fully meet PLOS Mental Health’s publication criteria as it currently stands. Therefore, we invite you to submit a revised version of the manuscript that addresses the points raised during the review process.

Please be aware that you are not obliged to implement all Reviewers` recommendations. If you choose not to accept some of them, please provide the justification in the rebuttal letter. **One of the crucial points is the Data Availability Policy of the PLOS Mental Health. **

We look forward to receiving your revised manuscript.

Kind regards,

Vitalii Klymchuk, Ph.D., D.Sc.

Academic Editor

PLOS Mental Health

Journal Requirements:

1. We have noticed that you have uploaded Supporting Information files, but you have not included a list of legends. Please add a full list of legends for your Supporting Information files after the references list. 

Additional Editor Comments (if provided):

Reviewers' comments:

Reviewer's Responses to Questions

**Comments to the Author**

1. Does this manuscript meet PLOS Mental Health’s publication criteria? Is the manuscript technically sound, and do the data support the conclusions? The manuscript must describe methodologically and ethically rigorous research with conclusions that are appropriately drawn based on the data presented.

Reviewer #1: Yes

Reviewer #2: Yes

2. Has the statistical analysis been performed appropriately and rigorously?

Reviewer #1: I don't know

Reviewer #2: Yes

3. Have the authors made all data underlying the findings in their manuscript fully available (please refer to the Data Availability Statement at the start of the manuscript PDF file)?

Reviewer #1: Yes

Reviewer #2: No

4. Is the manuscript presented in an intelligible fashion and written in standard English?

Reviewer #1: Yes

Reviewer #2: Yes

5. Review Comments to the Author

Reviewer #1: The article meets the PLOS Mental Health publication criteria. The manuscript presents a methodological and ethically rigorous investigation. And the data supports the conclusions. .

However, it is suggested to review the following observations:

1. Expand information on the procedures used for the application of the instruments and used for the development of the intervention

2. Describe the characteristics of informed consent and the procedure to request it.

3. It is necessary to expand the possible scope of the research. This would allow us to define the relevance of the publication of the manuscript.

4. In the conclusion, point out more specifically what the recommendations for future studies would be, above all, to improve assignment biases and to obtain more convincing results in the vocational and work area.

Reviewer #2: It would be great is the authors provided the link on the underlying data (that is accoding to the PLOS Data Availability Policy), and elaborated more in discussion and conclusions about policy implications of theirs findings.

6. PLOS authors have the option to publish the peer review history of their article (what does this mean?). If published, this will include your full peer review and any attached files.

**Do you want your identity to be public for this peer review?** For information about this choice, including consent withdrawal, please see our Privacy Policy.

Reviewer #1: No

Reviewer #2: No

---

## [Editor Report · Decision Letter 1]

22 Mar 2024

Does a specialized assessment improve vocational outcomes for people on sick leave with a suspected common mental disorder? Results from the Mental Health Assessment Study (MeHAS)

PMEN-D-23-00014R1

Dear Dr. Hoff,

We are pleased to inform you that your manuscript 'Does a specialized assessment improve vocational outcomes for people on sick leave with a suspected common mental disorder? Results from the Mental Health Assessment Study (MeHAS)' has been provisionally accepted for publication in PLOS Mental Health.

Best regards,

Vitalii Klymchuk, Ph.D., D.Sc.

Academic Editor

PLOS Mental Health
